# High-energy photoemission final states beyond the free-electron approximation

V. N. Strocov ●[1] ✉, L. L. Lev[1,2], F. Alarab ●[1], P. Constantinou ●[1], X. Wang[1], T. Schmitt ●[1], T. J. Z. Stock ●[3], L. Nicolaï ●[4], J. Očenášek ●[4] & J. Minár ●[4] ✉

Three-dimensional (3D) electronic band structure is fundamental for understanding a vast diversity of physical phenomena in solid-state systems, including topological phases, interlayer interactions in van der Waals materials, dimensionality-driven phase transitions, etc. Interpretation of ARPES data in terms of 3D electron dispersions is commonly based on the free-electron approximation for the photoemission final states. Our soft-X-ray ARPES data on Ag metal reveals, however, that even at high excitation energies the final states can be a way more complex, incorporating several Bloch waves with different out-of-plane momenta. Such multiband final states manifest themselves as a complex structure and added broadening of the spectral peaks from 3D electron states. We analyse the origins of this phenomenon, and trace it to other materials such as Si and GaN. Our findings are essential for accurate determination of the 3D band structure over a wide range of materials and excitation energies in the ARPES experiment.

Knowledge of electronic band structure resolved in three-dimensional (3D) electron momentum (**k**) is fundamental for understanding a vast diversity of physical phenomena in crystalline solid-state systems. Recently, the interest in 3D band structure has been boosted due to its essential role in topological phases such as Weyl semimetals (see, for example, refs. [1,2]) characterised by 3D cones of linear electron dispersion. These phases extend to high-fold chiral fermions[3,4] featuring high-dimensional degeneracies such as the Hopf links, nodal lines, chains and knots in 3D **k**-space (see the reviews[5–8] and the references therein). Less straightforward but equally important implications of the 3D band structure include, for example, interlayer interaction and 3D charge-density waves in van der Waals materials[9–11], formation of quantum-well states at interfaces and heterostructures[12–16], minibands in semiconductor superlattices[17], **k**-dependent electron-phonon interactions[18], dimensionality-driven phase transitions[19,20], 3D quantum Hall effect[21], and many more properties of solid-state systems.

Angle-resolved photoelectron spectroscopy (ARPES) is the unique technique that allows resolution of electronic band structure in **k**-space. The recent push of this technique to high excitation energies in the soft- and hard-X-ray photon energy (*hν*) regions has advanced its spectroscopic abilities from the conventional surface science to the electronic states deep in the bulk, buried interfaces and heterostructures, and diluted impurity systems (see the recent reviews[22–26] and the references therein). The main advantage of high photoelectron energies is an increase of the photoelectron mean free path ($\lambda_{PE}$) to a few nanometres and more[27]. Crucial for the experimental determination of 3D band structure, the increase of $\lambda_{PE}$ translates, via the Heisenberg uncertainty principle, to sharpening of the intrinsic resolution of the ARPES experiment in the out-of-plane momentum ($k_z$). This resolution ($\Delta k_z$) is defined, neglecting the surface effects[28–30] less significant at high energies, by Fourier transform of the exponentially decaying photoelectron wavefunction as $\Delta k_z = \lambda_{PE}^{-1}$[31]. The sharp $\Delta k_z$ underlies the applications of high-energy ARPES for accurate determination of the electronic band structure resolved in 3D **k**-space as illustrated by many works referred to in the above reviews.

In contrast to the in-plane momentum $\mathbf{k}_{//} = (k_x, k_y)$, conserved in the photoemission (PE) process because of the in-plane periodicity of the system, the $k_z$ component is distorted upon the photoelectron escape from the crystal to vacuum. It can however be reconstructed based on its conservation in the photoexcitation process in the bulk

[1]Swiss Light Source, Paul Scherrer Institute, 5232 Villigen-PSI, Switzerland. [2]Moscow Institute of Physics and Technology, 141701 Dolgoprudny, Russia. [3]London Centre for Nanotechnology, University College London, London WC1H 0AH, UK. [4]University of West Bohemia, New Technologies Research Centre, 301 00 Plzeň, Czech Republic. ✉e-mail: vladimir.strocov@psi.ch; jminar@ntc.zcu.cz

(corrected for the photon momentum $\mathbf{p}_{h\nu}$) if the final-state $k_z$ back in the crystal is known. Conventionally, the final-sate dispersion is modelled within the free-electron (FE) approximation, where the out-of-plane momentum of the photoelectron in the crystal ($K_z$) is found as $K_z = \frac{\sqrt{2m}}{\hbar}\sqrt{E_k - \frac{\hbar^2}{2m}K_{//}^2 - V_0}$, with $E_k$ and $\mathbf{K}_{//}$ being the photoelectron kinetic energy and in-plane momentum, respectively, $m$ the free-electron mass, and $V_0$ the inner potential. Somewhat stretching this formula, an energy dependence of the dynamic exchange-correlation[32,33] can be accommodated via an energy-dependent $V_0$. Importantly, the FE approximation implies that the final-state wave-function is one single plane wave $e^{i\mathbf{K}\mathbf{r}}$. This description can incorporate the finite $\lambda_{PE}$ via complex $K_z$ where Im$K_z$ is equal to $(2\lambda_{PE})^{-1}$.

It has since long been realised that at low excitation energies used in the conventional VUV-ARPES the FE approximation may in many cases fail even for metals[34–37] and all the more for semiconductors[38] and more complex materials such as transition metal dichalcogenides[39–41]. Apart from non-parabolic band dispersions, a remarkable aspect of the non-FE final states is that they can incorporate multiple bands having different $k_z$. A scheme of such multiband final states (MBFSs) is shown in Fig. 1. Their wavefunction (a, top) is a superposition of multiple (here a pair of) Bloch waves (decaying into the crystal within $\lambda_{PE}$) having different $k_z$ (with Im$k_z$ describing the decay). The corresponding out-of-plane band structure $E(k_z)$ (b) features multiple dispersion branches. In this case one initial-state band at the binding energy $E_B^0$ will yield, via the $k_z$ conservation, multiple ARPES peaks as a function of $E_k$ – in our scheme at $E_k^1$ and $E_k^2$ – where the FE approximation would predict one. Complementary, one final state at $E_k^0$ incorporates multiple bands, whose different $k_z$ will yield multiple peaks as a function of $E_B$ – in our scheme at $E_B^1$ and $E_B^2$ – from one single initial band. Depending on $\Delta k_z$ of the final-state bands (shown by shading) compared to their $k_z$ separation, these peaks either separate or merge into one broader peak.

For high-energy ARPES, however, the relevance of the FE approximation is commonly taken for granted. Being quintessential for 3D band mapping, this assumption is based on a physically appealing argument that if $E_k$ of photoelectrons much exceeds modulations of the crystal potential $V(\mathbf{r})$, they can be considered as free particles. Here, we analyse soft-X-ray ARPES data on Ag the metal and demonstrate that even at high excitation energies the complexity of the final states can go far beyond the FE picture. In particular, they

feature the MBFSs where the multiple $k_z$s manifest themselves as a complex structure or added broadening of the spectral peaks. Our analysis extends to GaN and Si the semiconductors. We theoretically demonstrate the origin of these non-trivial effects as resulting from hybridization of plane waves on the modulated $V(\mathbf{r})$, and elucidate how they should be taken into account for accurate determination of 3D valence-band dispersions in the high-energy ARPES experiment.

## Results

Figure 2 presents the Brillouin zone (BZ) of the fcc Ag (a) and the experimental out-of-plane cross-section of the Fermi surface (FS) in the ΓXW symmetry plane measured under variation of $h\nu$ (b). The indicated $K_z$ values, running through a sequence of the Γ and X points, were rendered from the $h\nu$ values assuming FE final states with $V_0 = 10$ eV. In-plane cross-sections measured at two $h\nu$ values, bringing $K_z$ to the Γ and X points, are presented in the two panels (c). In general, the experimental out-of-plane FS follows a pattern of repeating rounded contours characteristic of the states near the Fermi level ($E_F$) formed by the $sp$-band of Ag. This pattern goes along with our one-step ARPES calculations (e) where FE-like final states were used. Surprisingly, however, a closer look at the experimental FS reveals significant deviations: (1) Multiple FS contours, offset in $K_z$, can be resolved in some ($k_x$,$K_z$) regions such as those marked by magenta arrows. The corresponding multiple dispersions of the spectral peaks coming from the $sp$-band are apparent, for example, in the ARPES image measured at $h\nu = 997$ eV (d, top) and the corresponding momentum-distribution curve as a function of $k_x$ at $E_F$ ($k_x$-MDC, yellow line). This multiple-dispersion pattern contrasts to the clean dispersions at $h\nu = 894$ eV (d, bottom), for example. The first idea to explain the observed multiple dispersions might be formation of out-of-plane charge-density waves. However, no such phenomena have ever been reported for the unreconstructed Ag(100) surface; if even such waves had formed, the $K_z$ splitting of the multiple dispersion would have been independent of ($k_x$,$K_z$), which is obviously not the case. As we outlined in the Introduction, such multiple spectral structures actually identify the MBFSs incorporating multiple bands with different $k_z$ (the situation sketched on Fig. 1b, left) a phenomenon beyond the conventional FE-like final states implying one single band with one $k_z$. In our case the separation of the $k_z$s in these MBFSs is larger than the intrinsic $\Delta k_z$ (according to the $\lambda_{PE}$ values from the TPP-2M formula, varying from -0.15 Å$^{-1}$ at 300 eV to 0.056 Å$^{-1}$ at 1300 eV); (2) The second type of deviations from

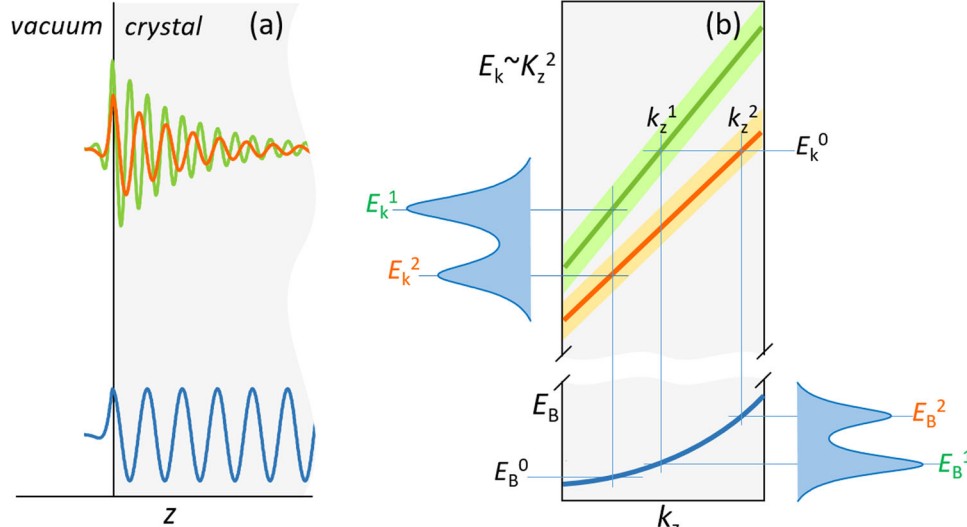

**Fig. 1 | Scheme of multiband final states (MBFSs). (a**, top) Multiple (here a pair of) final-state Bloch waves and (bottom) the initial-state one; **b** The corresponding out-of-plane band structure features multiple final-state bands, which yield

multiple ARPES peaks from one initial band. The shading shows $\Delta k_z$ of the final-state bands.

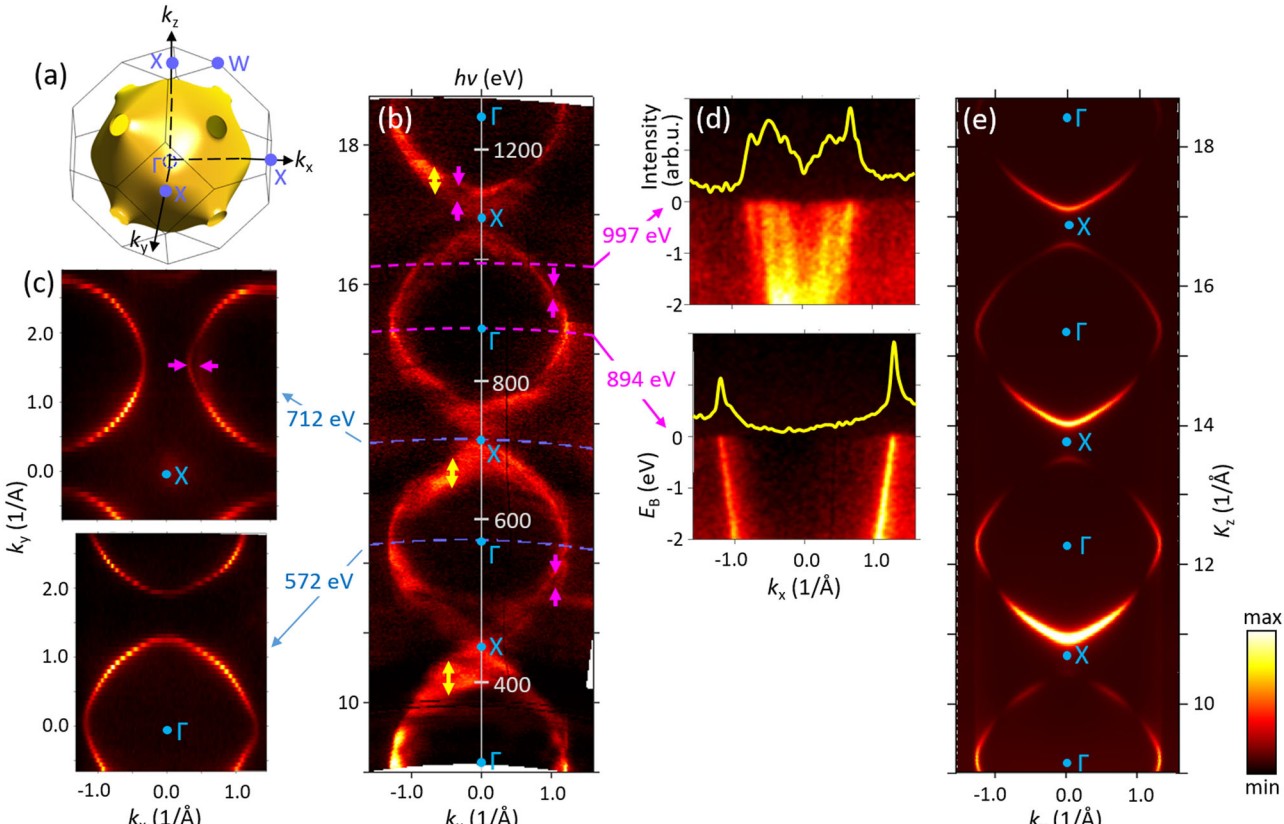

**Fig. 2 | FS cross-sections for Ag(100).** Theoretical FS (**a**), its experimental out-of-plane cross-section (**b**), and two in-plane cross-sections (**c**) measured at the indicated $h\nu$ values, bringing $K_z$ to the $\Gamma$ and X points (lower and upper panels, respectively). Replicas and broadening of the FS contours in certain $(k_x, K_z)$ regions (such as those marked by magenta and yellow arrows, respectively) manifest MBFSs. These effects are particularly clear in the ARPES image and $k_x$-MDC at $h\nu = 997$ eV (**d**, top) in contrast to those at $h\nu = 894$ eV (bottom). These effects are beyond the one-step ARPES calculations with FE-like final states (**e**).

the FE final states, seen in the out-of-plane FS (b), is a notable spectral intensity spreading into the X points where the *sp*-band is unoccupied. Furthermore, broadening of the FS contours in $k_z$ irregularly varies through **k**-space, and in some $(k_x, K_z)$ regions (such as those marked by yellow arrows) can be excessive. These two effects are also caused by the MBFSs, but in this case the $k_z$s are separated less than $\Delta k_z$. We note that in the extremes of the $E(k_z)$ dispersion ($dk_x/dk_z = 0$ in the out-of-plane FS) the MBFSs have only a second-order effect on the ARPES structure; however, even in this situation a large enough $k_z$ separation within the MBFSs can cause multiple FS contours, as seen in the in-plane FS map measured at $h\nu = 712$ eV (c, magenta arrow). Obviously, the MBFS effects are not reproduced by the ARPES calculations (e) employing FE final states. Although presently on a qualitative level, these effects are reproduced by our one-step ARPES calculations—see Supplementary Information (SI)—where the final states are treated within the multiple-scattering formalism, naturally incorporating the non-FE effects including the MBFSs.

In Fig. 3, the theoretical $E(\mathbf{k})$ along the $\Gamma$X direction (a) is compared with the experimental out-of-plane band dispersions $E(k_z)$ at $k_x = 0$ (b) and the in-plane $E(\mathbf{k}_{//})$ images (c) measured at $K_z$ running through the successive $\Gamma$ points (energies as binding energies $E_b$ relative to $E_F$). Again, the gross structures of the experimental ARPES intensity follow the expected periodic pattern with the *sp*-band crossing $E_F$ as reproduced by our one-step ARPES calculations in (e) with the FE-like final states. We see, however, replicas and anomalous broadening of the *sp*-band (such as marked by magenta arrows) as well as significant spectral intensity around the X point. These anomalies appear most clearly in the zoom-in of the *sp*-band and the $K_z$-MDC at $E_F$ (d, yellow line) where we observe a complex multi-peak structure of the spectral intensity around

the X point. Again, these effects are manifestations of the MBFSs (as sketched on Fig. 1b, right) with the ARPES dispersions originating from the individual final-state bands marked by the magenta arrows. Again, they are absent in the ARPES calculations employing FE final states (e) but are qualitatively reproduced upon inclusion of multiple-scattering final states, see Supplementary Fig. 1. The MBFS effects could not be observed in the first soft-X-ray study on Ag(100) focused on the 3*d* states[42] because the smaller $k_z$ dispersion of these states compared to the *sp* ones could not provide sufficient separation of the spectral peaks from the different bands in the MBFS. We note in passing that the experimental 3*d* states appear in ~1 eV below the LDA-DFT energies; such an energy shift, already noticed for Cu, is a pronounced self-energy effect due to non-local exchange interaction of the 3*d* electrons strongly localised in the core region[43].

## Discussion
### Origin of the MBFSs
By definition, a FE-like final state in the crystal is one single plane wave $e^{i(\mathbf{k}+\mathbf{G})\mathbf{r}}$ which matches the outgoing photoelectron plane wave. In the whole multitude of bands, formally available under $E_k$ and $\mathbf{K}_{//}$ conservation, this plane wave corresponds to one single band that we will refer to as primary, relaying Mahan's primary-photoemission cones[44]. All other bands in the multitude give strictly zero contribution to the photocurrent. We will be calling them secondary, relaying Mahan's secondary cones. The MBFS effects, observed in our ARPES data, indicate that the corresponding final states may include, for given $E_k$ and $\mathbf{K}_{//}$, several bands with different $k_z$ giving comparable contributions to the ARPES intensity. These effects obviously fall beyond the FE-like picture. As the first-principles calculations can not yet exhaustively

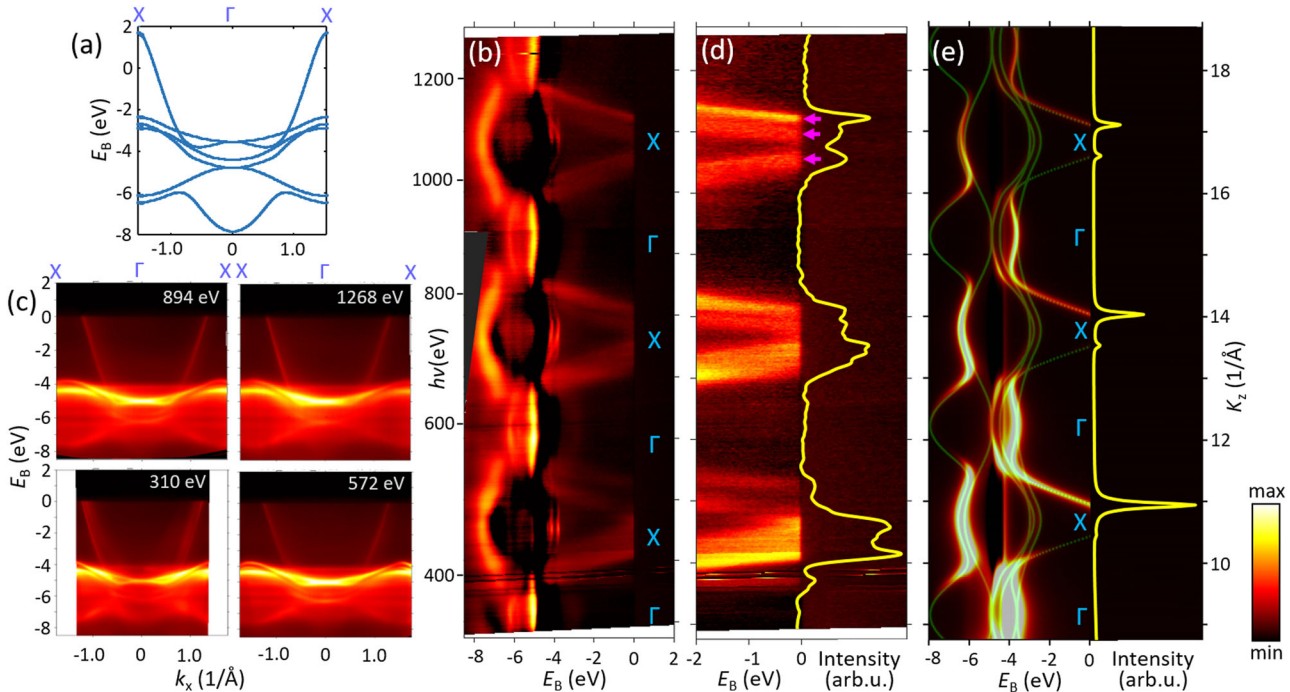

**Fig. 3 | Band dispersions along the ΓX direction for Ag(100).** Theoretical $E(\mathbf{k})$ (**a**) compared with the experimental out-of-plane ARPES dispersions at $k_x = 0$ (**b**, the spectral intensity represented in logarithmic scale) and (**c**) in-plane dispersions for the indicated $h\nu$ values, bringing $K_z$ to the successive Γ points. A zoom-in of the *sp*-band (**d**) shows its replicas and added broadening (such as marked by magenta arrows) most evident in the $K_z$-MDC at $E_F$ (yellow line) as multiple and broadened spectral peaks, manifesting the MBFSs. These effects are beyond the one-step calculations of the ARPES intensity and $K_z$-MDC with FE-like final states (**e**).

describe our experimental results, we will analyse the MBFS effects based on insightful model calculations.

The non-FE effects in the final states, in particular their multiband composition, is certainly a phenomenon not new for low-energy ARPES. They have been studied experimentally and theoretically for 3D bulk band dispersions in various materials including Cu[34,35], Mg[37] and even Al the paradigm FE metal[14,45], semiconductors[38], various transition metal dichalcogenides[39–41] as well as surface states, in particular for the Al(100) and (111) surfaces[36]. However, it is intriguing to observe such effects in our soft-X-ray energy range. Why do they appear in spite of the fact that the photoelectron $E_k$ is overwhelmingly large compared to the $V(\mathbf{r})$ modulations?

We will now build a physically appealing picture of the non-FE effects in the PE final states using their standard treatment as the time-reversed LEED states[28]. They are superpositions of damped Bloch waves $\phi_\mathbf{k}(\mathbf{r})$ with complex $k_z$, whose imaginary part $\text{Im}k_z$ represents the (1) inelastic electron scattering, described by a constant optical potential $V_i$ (imaginary part of the self-energy), and (2) elastic scattering off the crystal potential[46–49]. The amplitudes $T_\mathbf{k}$ of these $\phi_\mathbf{k}(\mathbf{r})$, which determine their activity in LEED (and, as we will see, also in ARPES) were determined within the matching approach of the dynamic theory of LEED[17,41,46,47,50,51]. In this approach, the electron wavefunction in the vacuum half-space (superposition of the incident plane wave $e^{i\mathbf{K}_0\mathbf{r}}$ and all diffracted ones $e^{i(\mathbf{K}+\mathbf{g})\mathbf{r}}$, **g** being the surface reciprocal vectors) is matched, at the crystal surface, to that in the crystal half-space (superposition of $\phi_\mathbf{k}(\mathbf{r})$ satisfying the surface-parallel momentum conservation $\mathbf{k}_{//}=\mathbf{K}_{//}+\mathbf{g}$). For details of the matching approach see the SI. The underlying complex band structure calculations utilised the empirical-pseudopotential scheme, where $\phi_\mathbf{k}(\mathbf{r})$ are formed by hybridization of plane waves $e^{i(\mathbf{k}+\mathbf{G})\mathbf{r}}$, **G** being 3D reciprocal-lattice vectors. The Fourier components $V_{\Delta\mathbf{K}} = \langle e^{i(\mathbf{k}+\mathbf{G})\mathbf{r}} | V(\mathbf{r}) | e^{i(\mathbf{k}+\mathbf{G}')\mathbf{r}} \rangle$ of the local pseudopotential $V(\mathbf{r})$ were adjustable parameters.

We start from the ideal FE case, where $V(\mathbf{r})$ is constant and equal to $V_0$ (so-called empty lattice). The corresponding calculations are plotted in Fig. 4a as the $E(\text{Re}k_z)$ bands (the complementary $E(\text{Im}k_z)$ bands are shown in the Supplementary Fig. 3). Due to the absence of hybridization between the plane waves in the empty-lattice case, each $\phi_\mathbf{k}(\mathbf{r})$ contains one single plane wave corresponding to a certain **G** vector. Typical of high energies, we observe a dense multitude of bands brought in by an immense number of all **G** vectors falling into our energy region. Starting from the ultimate $V_0 = 0$ case, when the vacuum half-space is identical to the crystal one, it is obvious that only one band will couple to the photoelectron plane wave in vacuum $e^{i\mathbf{K}\mathbf{r}}$ and thus be effective in the ARPES final state, specifically, only the primary band whose plane wave−in the context of LEED often called 'conducting' plane wave−has $\mathbf{k} + \mathbf{G}$ equal to the photoelectron **K**. The whole multitude of the secondary bands, whose plane wave's $\mathbf{k} + \mathbf{G}$ is different from **K**, will give no contribution to the photo-current. In our more general case $V(\mathbf{r}) = V_0$, the $k_z$ component of the photoelectron distorts upon its escape to vacuum, and the above momentum-equality condition to identify the conducting plane wave should be cast in terms of the in-plane components as $\mathbf{k}_{//} + \mathbf{G}_{//} = \mathbf{K}_{//}$.

In a formal language, these intuitive considerations can be expressed through the partial contributions of each $\phi_\mathbf{k}(\mathbf{r})$ into the total current absorbed in the sample in the LEED process, which are the so-called partial absorbed currents

$$I_\mathbf{k}^{\text{abs}} \propto V_i \cdot \int_0^\infty |T_\mathbf{k}\phi_\mathbf{k}(z)|^2 dz,$$

with the integration extending through the crystal half-space[34,35,40]. Importantly in the ARPES context, the $I_\mathbf{k}^{\text{abs}}$ values multiplied by the PE

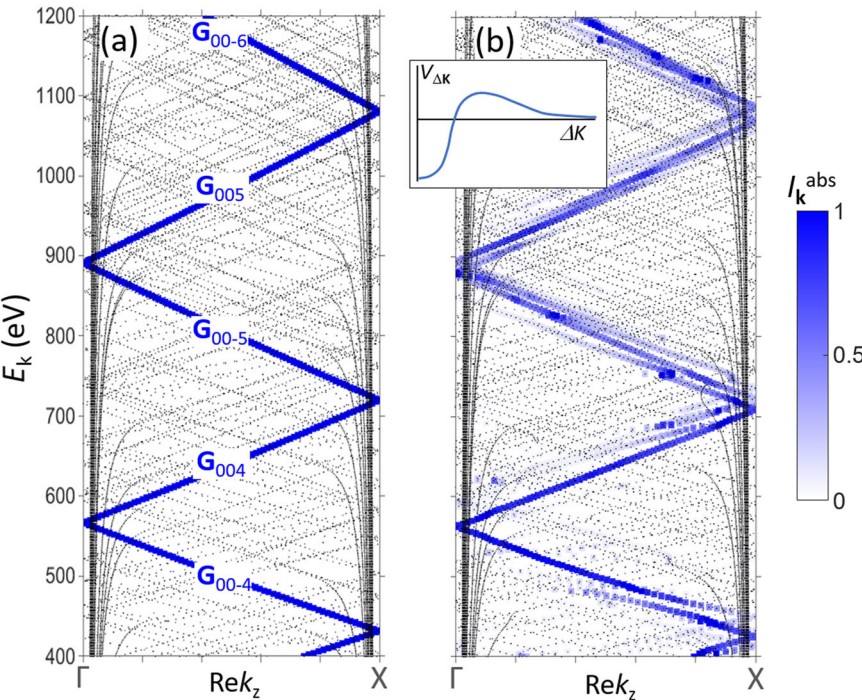

**Fig. 4 | Band structure of the final-state Bloch waves $E(\mathrm{Re}k_z)$ in a model fcc crystal along the $\Gamma$X direction. a** The empty-lattice case $V(\mathbf{r}) = V_0$ and (**b**) a realistic case of spatially modulated pseudopotential, whose typical $V_{\Delta \mathbf{K}}$ is sketched in the insert. The dense multitude of bands is formed by an immense number of **G** vectors falling into our high-energy region. The contributions of each band into the total photocurrent are quantified by $I_\mathbf{k}^{abs}$ (blue colorscale). Whereas in the first case the photocurrent emanates from one single FE band (marked with the corresponding **G** vectors), in the second case it may distribute over a few bands alongside the FE dispersion, which form a MBFS incorporating a few $k_z$s.

matrix element $M_{fi}$ define the partial photocurrents

$$I_\mathbf{k}^{PE} \propto \frac{1}{V_i} \cdot |M_{fi}|^2 \cdot I_\mathbf{k}^{abs}$$

emanating from the individual $\phi_\mathbf{k}(\mathbf{r})$ in the MBFS[34] (neglecting the cross-terms in the total absorbed and PE currents, see the SI). This means that the activities of the individual bands in the LEED and PE processes are parallel. In Fig. 4a the calculated $I_\mathbf{k}^{abs}$ are marked in blue colorscale. In our empty-lattice case, as expected, $I_\mathbf{k}^{abs}$ is equal to 1 for the primary (in the LEED context often called conducting) band and strictly zero for all other ones, realising the ideal FE final state containing one single plane wave. In Mahan's language, only the primary-cone PE is active in the ideal FE case.

We will now introduce spatial modulations of $V(\mathbf{r})$ as expressed by $V_{\Delta \mathbf{K}}$ for non-zero $\Delta \mathbf{K}$. The plane waves start to hybridise through the $V_{\Delta \mathbf{K}}$ matrix elements, and each $\phi_\mathbf{k}(\mathbf{r})$ becomes a superposition of a few plane waves as $\phi_\mathbf{k}(\mathbf{r}) = \Sigma_\mathbf{G} C_\mathbf{G} e^{i(\mathbf{k}+\mathbf{G})\mathbf{r}}$. In this case not only one but several $\phi_\mathbf{k}(\mathbf{r})$ can acquire a certain admixture of the $\mathbf{k}_{//} + \mathbf{G}_{//} = \mathbf{K}_{//}$ conducting plane wave− in the formal language, their $I_\mathbf{k}^{abs}$ becomes non-zero−and give a certain contribution to the total photocurrent. Our model calculations for this case are sketched in Fig. 4b. The ARPES final state appears multiband in a sense that it consists of several $\phi_\mathbf{k}(\mathbf{r})$ with different $k_z$ (typically alongside the primary band) which give comparable contributions to the total ARPES signal as quantified by the corresponding $I_\mathbf{k}^{abs}$. In Mahan's language, the qualitative distinction between the primary- and secondary-cone PE dissolves. Correspondingly, the ARPES spectra will show up several peaks corresponding to different $k_z$ or, if the separation of these $k_z$s is smaller than the intrinsic $\Delta k_z$, added broadening of the spectral peaks. This is exactly what we have just seen in our ARPES data on Ag(100). We note in passing that on the qualitative level the bands contributing to the photocurrent can be easily

identified based on the Fourier expansion of their $\phi_\mathbf{k}(\mathbf{r})$ which should have a substantial weight of the $\mathbf{k}_{//} + \mathbf{G}_{//} = \mathbf{K}_{//}$ conducting plane wave[52].

Whereas for the sake of physical insight we have intentionally simplified the above picture, the exact treatment of the MBFSs based on the matching approach of LEED has been developed in a series of previous works albeit limited to relatively low final-state energies[34,37,40,41]. Finally, we note that the MBFS phenomenon can also be understood within the simplified three-step model of PE splitted into the photo-excitation, photoelectron transport out of the crystal, and its escape to vacuum. In this framework, the MBFSs can be viewed as resulting from multiple scattering of photoelectrons on their way out of the crystal that creates multiple Bloch-wave modes of the scattered wavefield.

Whereas the effects of MBFSs have already been established at low excitation energies, their survival in high-energy ARPES might seem puzzling. In a naive way of thinking, photoelectrons with energies much higher than the modulations of $V(\mathbf{r})$ should not feel them, recovering the FE case with one single $\phi_\mathbf{k}(\mathbf{r})$. However, $V_{\Delta \mathbf{K}}$ as the strength of hybridization between two plane waves depends, somewhat counter-intuitively, not on energy but rather on $\Delta \mathbf{K}$ between them. As sketched in the insert in Fig. 4b, $V_{\Delta \mathbf{K}}$ typically has its maximal negative value at $\Delta K = 0$ (which is the $V_0$), and with increase of $\Delta K$ sharply rises and then asymptotically vanishes. Importantly, however high the energy is, the multitude of the plane waves always contains pairs of those whose $\Delta K$ is small. The corresponding bands can be identified by close dispersions. For such pairs $V_{\Delta \mathbf{K}}$ is large, giving rise to their strong hybridization. Importantly, all bands hybridising with the $\mathbf{k}_{//} + \mathbf{G}_{//} = \mathbf{K}_{//}$ plane wave will receive non-zero $I_\mathbf{k}^{abs}$ and thus contribute to the total photocurrent, as shown in Fig. 4b. This forms the MBFSs that should survive even at high energies.

## Effect of MBFSs on the spectral structure

We will now follow in more detail how the MBFSs affect the ARPES spectra. As an example, we will analyse the experimental $K_z$-MDC from

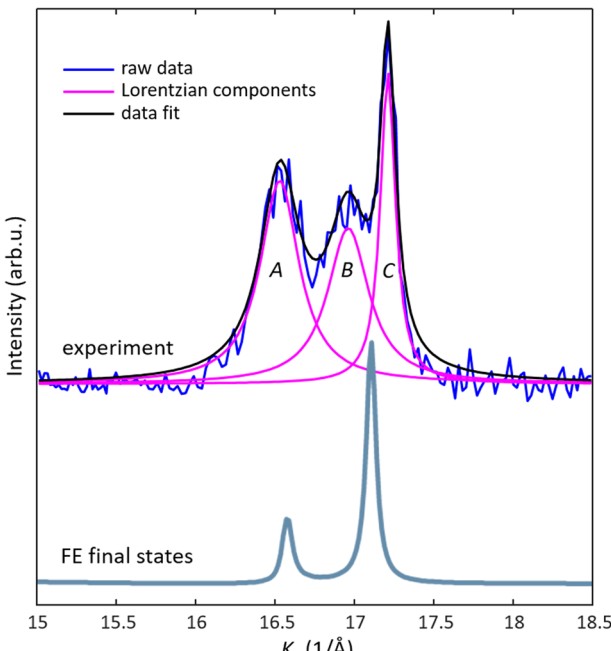

**Fig. 5 | $K_z$-MDCs at $E_F$ in the $h\nu$ region around 1100 eV (vicinity of the X point).** (top) Experimental one extracted from Fig. 3d compared with (bottom) the one from Fig. 3e calculated with FE final states. The complex structure of the experimental $K_z$-MDC, featuring three peaks with different broadening, is caused by the MBFSs.

Fig. 3d in the region of the X point at $h\nu$ ~ 1100 eV, reproduced on top of Fig. 5 (with the linear background subtracted). As illustrated by the corresponding $K_z$-MDC from Fig. 3e calculated within the FE approximation (bottom curve) we might expect to observe here two Lorentzian peaks, placed symmetrically around the X point and broadened by the same intrinsic $\Delta k_z$. However, the experimental $K_z$-MDC shows three distinct peaks $A$-$C$, with the peak $B$ coming from a final-state band falling beyond the FE approximation. Moreover, Lorentzian fitting of the peaks finds that whereas the peak $C$ has a relatively small width of $0.11\,\text{Å}^{-1}$, the widths of the peaks $A$ and $B$ are more than twice larger, $0.30$ and $0.32\,\text{Å}^{-1}$, respectively. The picture of MBFSs neatly explains this observation, suggesting that whereas the peak $C$ is formed by a final state having one dominant $k_z$ contribution, and the peaks $A$ and $B$ by final states incorporating a multitude of $k_z$ separated less than $\Delta k_z$. Whereas it is generally believed that the intrinsic broadening of the ARPES peaks in $k_z$ is determined exclusively by finite $\lambda_{PE}$ the photoelectron mean free path, our example demonstrates that the multiband final-state composition may not only create additional spectral peaks but also be an important factor of their broadening additional to $\lambda_{PE}$.

Intriguingly, however, we note that even the narrowest peak $C$ is almost twice broader than $\Delta k_z$ ~ $0.065\,\text{Å}^{-1}$ expected from $\lambda_{PE}$ ~ 15.5 Å suggested by the TPP-2M formula[53] well-established in XPS and Auger electron spectroscopy. One explanation might be that already the peak $C$ would incorporate multiple final-state bands with smaller $k_z$ separation compared to other two peaks. Another explanation would trace back to quasielastic electron-electron or electron-phonon scattering, which would increase with energy owing to the increase of the phase space available for such scattering.[54] Altering **k** of photoelectrons, it should destroy the coherence of photoelectrons and thus reduce $\lambda_{PE}$ as reflected in the observed broader $\Delta k_z$. At the same time, the quasielastic scattering should have only a little effect on attenuation of the **k**-integrated signal of the core-level or intrinsically incoherent Auger electrons. In other words, the effective $\lambda_{PE}$ in ARPES

should be smaller than that in XPS/Auger spectroscopy, described by the TPP-2M and related formalisms. Such intriguing fundamental physics certainly deserves further investigation.

## MBFS phenomena through various materials

The phenomenon of MBFSs surviving at high excitation energies is certainly not restricted to Ag only and, strengthening with the strength of $V(\mathbf{r})$ modulations, should be fairly general over various materials. Even for Al the paradigm FE metal, astonishingly, such MBFSs can be detected at least up to excitation energies of a few hundreds of eV[14,45]. Quite commonly the MBFS effects at high energies are observed in van-der-Waals materials such as $MoTe_2$[55], which should be connected with a large modulation of $V(\mathbf{r})$ across the van-der-Waals gap.

Another vivid example of the MBFS effects is the soft-X-ray ARPES data for GaN presented in Fig. 6 (compiled from the previously published results on the GaN(1000) layer in AlN/GaN heterostructures[13]). The panel (a) shows the ARPES spectral structure plot as a function of $K_z$ along the out-of-plane ΓA direction of the bulk BZ, which is expected from the DFT valence bands and FE final states with $V_0 = 5$ eV (energies relative to the valence-band maximum $E_{VBM}$). With the non-symmorphic space group of bulk GaN, the ARPES dispersions allowed by the dipole selection rules (though somewhat relaxed due to the presence of the surface[30] and band bending in GaN breaking the 3D periodicity) are marked bold. The panels (b,c) present the experimental out-of-plane ARPES dispersions measured at $k_x$ in two formally equivalent points, $\bar{\Gamma}_0$ in the first surface BZ (normal emission) and $\bar{\Gamma}_1$ in the second one. Their dramatic deviations from the predictions of the FE approximation reveal pronounced MBFS effects. Their strength much exceeds those in Ag because of weaker electron screening of the atomic potential and thus sharper modulations of $V(\mathbf{r})$ in the covalent GaN compared to the metallic Ag. The individual dispersion branches (marked by arrows at their top) are separated in $K_z$ more than the intrinsic $\Delta k_z$ broadening. In their multitude, one can identify the one that can be associated with the primary-cone PE (bold arrows) and is not necessarily the strongest. Remarkably, for the same initial-state $E(\mathbf{k})$ the ARPES dispersions measured at the formally equivalent $\bar{\Gamma}_0$ and $\bar{\Gamma}_1$ appear totally different. This fact is attributed to different final-state bands selected from the whole continuum of unoccupied states available for given final-state energy and $\mathbf{K}_{//}$; these bands are identified by their leading plane-wave component $\mathbf{k}_{//} + \mathbf{G}_{//}$ being equal to $\mathbf{K}_{//}$ of the detected photoelectrons, where the latter changes between the surface BZs[35].

The high-energy final states in Si are a counter-example though. Figure 7a shows the ARPES spectral structure plot along the out-of-plane ΓX direction expected from the DFT-GGA calculated valence bands and FE final states with $V_0 = 12.6$ eV. Here, the dispersions of the split-orbit-split band in bold and pale represent the favourable selection rules in the $\bar{\Gamma}_0$ and $\bar{\Gamma}_1$ points of two surface BZs, respectively (P. Constantinou et al., unpublished). The panels (b,c) present the corresponding out-of-plane soft-X-ray ARPES data, measured on Si(100) thin film n-doped with As[56]. The arrows mark the allowed split-orbit-split band dispersions different in $\bar{\Gamma}_0$ and $\bar{\Gamma}_1$. Because of the covalent character of Si, one might again expect that the non-FE effects here would be comparable to those for GaN and in any case stronger than for the metallic Ag. Contrary to such expectations, however, the experimental dispersions in both BZs do not show any clear signatures of the MBFSs additional to the FE case (a), at least in the shown $(E_k, \mathbf{k})$ region. No such signatures can also be resolved in the experimental out-of-plane iso-$E_B$ contours (d). This fact is particularly surprising because at low excitation energies the MBFS effects in Si are profound[38]. At the moment we can not decipher any simple arguments that would relate the strength of the non-FE effects in the high-energy electron states to any basic electronic-structure parameters of various materials.

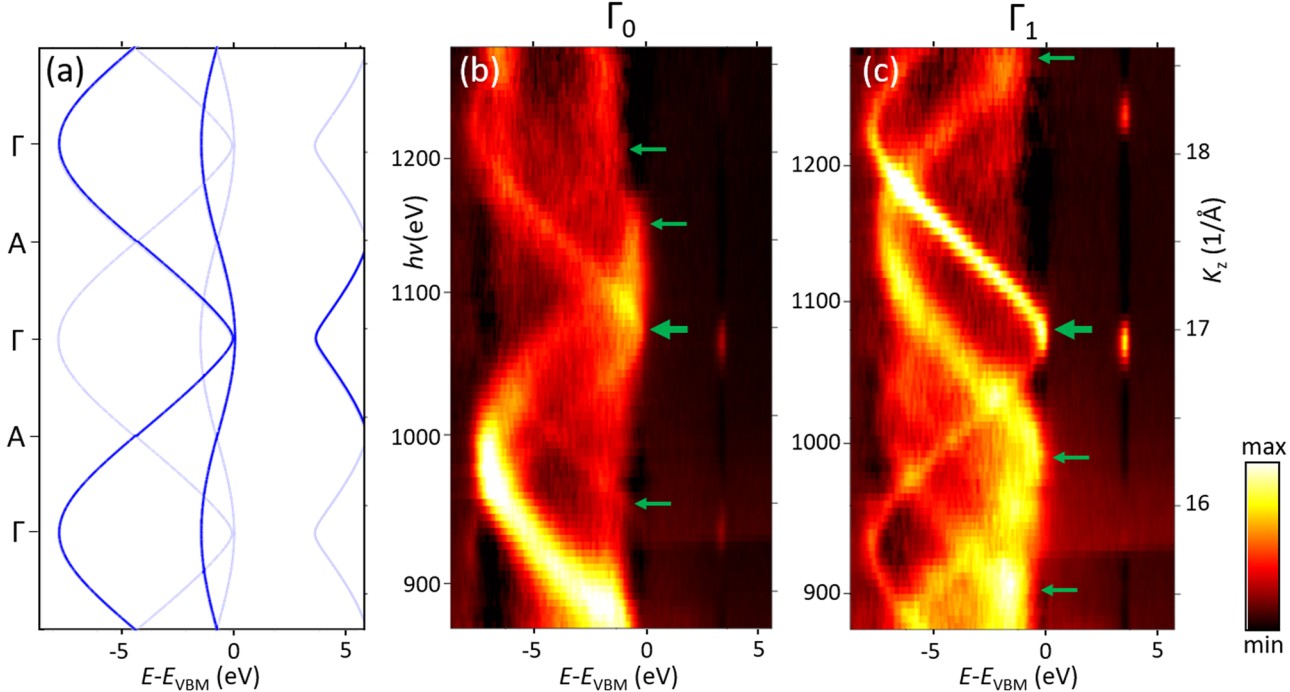

**Fig. 6 | Out-of-plane ARPES data for GaN(1000) along the ΓA direction.**
**a** Dispersions expected from the DFT valence bands and FE final states with $V_0 = 5$ eV. With the non-symmorphic space group of bulk GaN, the dispersions allowed by the dipole selection rules are shown bold; **b**, **c** Measured in the $\bar{\Gamma}_0$ and $\bar{\Gamma}_1$ points of two surface BZs (note slightly shifted $h\nu$ scales). The experiment clearly resolves individual final-state bands (marked by arrows) whose separation in $k_z$ is larger than the intrinsic $\Delta k_z$ broadening.

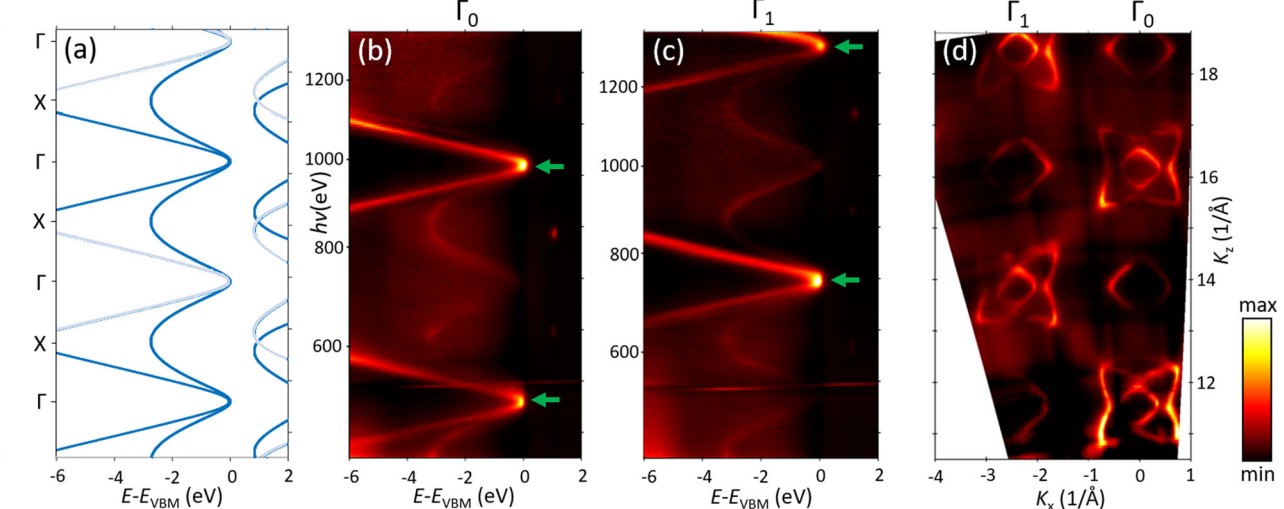

**Fig. 7 | Out-of-plane ARPES data for Si(100) along the ΓX direction. a** Dispersions expected from the DFT valence bands and FE final states with $V_0 = 12.6$ eV. The spin-orbit split band dispersions allowed by the selection rules in the $\bar{\Gamma}_0$ and $\bar{\Gamma}_1$ points of two surface BZs are shown bold and pale, respectively; **b**, **c** Measured in the $\bar{\Gamma}_0$ and $\bar{\Gamma}_1$ points. The arrows mark the allowed spin-orbit split band dispersions; (**d**) Iso-$E_B$ contours measured at 2 eV below $E_{VBM}$. No signatures of the MBFSs can be identified in these data.

## Non-FE effects beyond ARPES

The non-FE effects in high-energy electron states such as MBFS manifest themselves not only in the ARPES dispersions. Another manifestation will be the circular dichroism in the angular distribution of photoelectrons (CDAD) that necessitates that the final-state wavefunctions deviate from the free-electron plane waves[57–59]. The CDAD has indeed been observed already in the early soft-X-ray ARPES study on Ag(100)[42]. Another example is the orbital tomography of adsorbed molecules (see, for example, refs. 60–62) which takes advantage of the

Fourier relation between the angle distribution of photoelectrons and electron density of the valence electron orbitals. The non-FE effects introduce additional plane-wave components in the final states, calling for refinement of the straightforward Fourier-transform processing of the experimental data[62]. Beyond ARPES, the very fact of electron diffraction at crystalline surfaces identifies non-FE effects in the electron states in the crystal, because otherwise the incident electrons would upon entering the crystal follow the same FE wavefunction and thus would not reflect. The reflection high-energy electron diffraction

(RHEED) evidences that the non-FE effects survive even in the energy range of a few tens of keV, when $\Delta\mathbf{K}$ between the incident and diffracted plane waves is small and thus the corresponding $V_{\Delta\mathbf{K}}$ large. These considerations suggest that the MBFSs should survive even in hard-X-ray ARPES, waiting for a direct experimental observation.

Finally, we should point out that the coherent PE process underlying the ARPES experiment discussed above (as well as the orbital tomography) is fundamentally different to the essentially incoherent process of X-ray photoelectron diffraction (XPD) (see, for example, the reviews[63–65]). In the first case, all photoelectron emitters (atoms) throughout the crystal surface region within the depth $\lambda_{PE}$ are coherent—or entangled, in the modern quantum mechanics discourse—and emit a coherent photoelectron wavefield characterised by a well-defined $\mathbf{k}$. The resulting ARPES intensity as a function of $E_k$ and $\theta$ bears sharp structures reflecting, through the momentum conservation, the $\mathbf{k}$-resolved band structure of the valence states. In the XPD, other way around, the coherence between the emitters throughout the surface region is lost. This takes place, for example, for isolated impurity atoms, adsorbed molecules and localised core levels, where the initial-state wavefunctions at different atoms are decoupled from each other. Another case is when the coherence of photoelectrons is broken by thermal or defect scattering, or when the signal from certain valence-band states, like $d$-states, is integrated in energy[66,67]. The result is that each photoelectron emitter creates scattered waves within a sphere of the radius $\lambda_{PE}$, which interfere with each other incoherently with the waves emanating from another emitter. Typical of diffraction with a few interfering rays, the resulting XPD intensity distribution as a function of $E_k$ and $\theta$ is fairly smooth, and reflects the local atomic structure. With $E_k$ increasing into the hard-X-ray energy range, $\lambda_{PE}$ and thereby the number of coherently scattered waves increases. This forms sharp Kikuchi-like structures in the XPD angular distribution, reflecting the long-range atomic structure[65]. In any case, the XPD stays incoherent between the emitters. This fundamental difference between the coherent PE and incoherent XPD processes is stressed, for example, by the fact that in the first case the photoelectron angular distribution follows $\mathbf{p}_{h\nu}$, shifting with $h\nu$, and in the second case it is insensitive to $\mathbf{p}_{h\nu}$[19].

In conclusion, our analysis of extensive soft-X-ray ARPES data on the Ag metal has demonstrated that even at high excitation energies the PE final states may, intriguingly, in some energy and $\mathbf{k}$-space regions feature pronounced multiband composition beyond the conventional FE approximation. The corresponding Bloch waves have different $k_z$ momenta, typically alongside the FE dispersion, and give comparable contribution to the ARPES spectra. Using empirical-pseudopotential simulation of the final states, where these contributions were quantified as proportional to the partial current in each Bloch wave determined within the wavefunction-matching formalism of LEED, we have demonstrated that the MBFSs appear due to hybridization of plane waves through low-$\mathbf{K}$ components of the crystal potential. Depending on the $k_z$ separation of the individual Bloch waves, the MBFSs give rise to multiple ARPES peaks from 3D valence-band dispersions or become an important factor of their broadening in addition to the intrinsic $\Delta k_z$ broadening due to the finite $\lambda_{PE}$. From the first principles, these effects can be described by one-step ARPES calculations with the final states treated within the multiple-scattering or Bloch-wave approaches. Although our KKR-based calculations on Ag were able to qualitatively describe the experimental results, further theoretical effort is required to achieve a quantitative agreement at high excitation energies. Besides Ag, the MBFS phenomena are observed, for example, in previous soft-X-ray data on the covalent GaN and even Al, the paradigm FE metal. They are surprisingly weak, however, for the covalent Si. The MBFS phenomenon, typically strengthening with the sharpness of the crystal-potential modulations,

should be fairly general over a wide range of materials and excitation energies even into the hard-X-ray range.

## Methods

### Experiment

The experiments were performed at the soft-X-ray ARPES facility[68] installed at the high-resolution ADRESS beamline[69] of the Swiss Light Source, Paul Scherrer Institute, Switzerland. X-rays irradiated the sample with a flux of ~$10^{13}$ photons/s. Their grazing incidence on the sample with an angle of 20° increased the photoelectron yield by a factor of ~3 compared to the standard 45°.[68] A single crystal of Ag(100) (MaTecK) was cleaned by a few cycles of Ar ion sputtering/annealing. The LEED patterns showed the (1 × 1) surface structure without any reconstruction. The sample was cooled down to ~12 K in order to quench relaxation of $\mathbf{k}$-conservation due to thermal motion of the atoms[70], with the coherent spectral fraction enhanced by subtracting the angle-integrated spectrum scaled under the condition of non-negativity of the remaining spectral weight. The measurements were performed with $p$-polarised X-rays with the analyzer slit oriented in the X-ray incidence plane. The combined energy resolution varied from ~50 to 180 meV when going from $h\nu = 300$ to 1300 eV, which is about twice better than in the first soft-X-ray ARPES study on Ag(100)[42]. The FS maps were integrated over an $E_B$ window from −75 to 25 meV relative to $E_F$. Angular resolution of the analyzer PHOIBOS-150 was ~0.1°. Other relevant experimental details, including the conversion of $E_k$ and emission angle $\theta$ to $\mathbf{k}$, corrected for $\mathbf{p}_{h\nu}$, can be found elsewhere[68]. The data on GaN(1000) and Si(100) taken from the previous ARPES works were acquired under the same experimental conditions, but with the energy resolution relaxed to ~80 to 250 meV in the same $h\nu$ range. The absence of any surface reconstruction in our GaN(1000) samples had been evidenced by the RHEED images acquired during their MBE growth. The Si(100) samples had been etched in buffered HF solution in order to quench the dangling bonds[56] that ensured the formation of the (1 × 1) surface without any surface reconstructions as well.

### Calculations

In our simulations of the PE final states, we used an empirical local pseudopotential that allowed reduction of the secular equation on complex $k_z$ to an eigenvalue problem for a complex non-Hermitian matrix[17,47]. For the energy range of our simulation extending to 1200 eV, the basis set included all plane waves below an energy cutoff of 1800 eV. The inner potential $V_O$ was set to 10 eV, all $V_{\Delta\mathbf{K}}$ to 5 eV for $\Delta K^2 < 48$ and to zero for larger $\Delta K^2$, and $V_i$ to 5 eV. The $l_k^{abs}$ values of the Bloch waves have been calculated within the matching approach of LEED. The accuracy of the calculations was controlled via the current conservation generalised for non-zero $V_i$ on the crystal side. For our qualitative analysis of the final states, no attempt has been made to fit these parameters to our particular case. For an outline of the matching approach and further computational details see the SI.

The first-principles ARPES calculations were performed using the SPR-KKR package[71] relying on the multiple scattering theory using the Korringa-Kohn-Rostoker (KKR) method. The ground-state properties of the Ag(001) surface were derived from density-functional-theory (DFT) calculations within the local-density approximation (LDA) carried out with full potential. The ARPES spectra were calculated within the one-step model of PE in the spin-density-matrix formulation[72] taking into account all aspects of the PE process for the actual experiment including $\mathbf{p}_{h\nu}$, matrix elements and final states constructed as the time-reversed LEED states. Taking into advantage the predominance of forward scattering at $E_k$ above ~400 eV (D. Sébilleau, S. Tricot & A. Koide, unpublished, see the Supplementary Fig. 2) the calculations used the single-site scattering approximation. The final-state damping was described

via constant $V_i = 3\,\text{eV}$ set to reproduce $\lambda_{PE} = 10.2\,\text{Å}$ at $E_k = 600\,\text{eV}$ given by the TPP-2M formula[53]. The main paper presents the results obtained with FE final states, and the effects of multiple-scattering final states and various computational approximations are discussed in the SI. We note that the agreement with the experiment has anyway stayed on a qualitative level. The accuracy of our calculations might be limited because (1) the angular-momentum cutoff $l_{max} = 5$ was yet insufficient for high energies; (2) the total-energy minimization used to generate the self-consistent potential might be quite insensitive to its high-frequency Fourier components, critically affecting the high-energy electron states.

## Data availability

The raw and derived data presented are available from the corresponding authors upon a reasonable request.

## Code availability

The codes used for the theoretical calculations and data processing are available from the corresponding authors upon a reasonable request.

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

## Acknowledgements

V.N.S. thanks E.E. Krasovskii for illuminating discussions and critical reading of the manuscript, J.H. Dil for valuable exchange on physics of XPD, and I. O. Maiboroda for the structural information on the GaN samples. J.M. is grateful to D. Sébilleau, S. Tricot and A. Koide for sharing their scattering-amplitude calculations. The authors thank N.J. Curson and S.R. Schofield for giving access to Si samples prepared at University College London. J.M. and L.N. acknowledge the support of the Czech Ministry of Education, Youth and Sports via the grant CEDAMNF CZ.02.1.01/0.0/0.0/15_003/0000358 and the support from GACR Project No. 2018725S. L.L.L. acknowledges the financial support from the Ministry of Science and Higher Education of the Russian Federation, grant #075-11-2021-086. F.A. acknowledges the financial support from the Swiss National Science Foundation within the grant 200020B_188709. T.J.Z.S. acknowledges the financial support of the Engineering and Physical Sciences Research Council (grants nos. EP/R034540/1, EP/W000520/1), and Innovate UK (grant no. 75574).

## Author contributions

V.N.S. and J.M. conceived the SX-ARPES experiment at the Swiss Light Source. V.N.S., L.L.L., F.A., P.C and L.N. performed the experiment supported by X.W. and T.S. T.J.Z.S. fabricated the thin-film Si samples. V.N.S. processed and interpreted the data, and performed computational simulation of the final states supported by P.C. J.M. performed the first-principles ARPES calculations. V.N.S. wrote the manuscript with contributions from J.M., L.L.L., P.C., T.J.Z.S. and J.O. All authors discussed the results, interpretations, and scientific concepts.

## Competing interests

The authors declare no competing interests.
