## [Peer Review File · Nature Communications]

REVIEWER COMMENTS

Reviewer #1 (Remarks to the Author):

This report by Strocov et al. is an important work leading to an accurate understanding and precise analysis of photoelectron spectroscopy. This is because the authors attempt a detailed comparison with the free-electron approximate final state and the more realistic pseudopotential approximate final state, based on several experimental examples. Since the free electron approximation is only an approximation method, it is natural that the discrepancy between the actual and the approximation will eventually be exposed once a precise discussion begins, even in the high-energy region. In this sense, the title 'Is the approximation valid?' at first glance seems to already provide a 'no' answer, but it was interesting in that it gave both examples where the approximation breaks down (Ag and GaN) and where it does not (Si).

In recommending this manuscript to your journal, I would like to make the following three comments. All of these points are of concern to me as a first-time reader, and I hope that the authors will use them to improve this manuscript.

1) relation of mean free path and k_z dispersion resolution

P2-l48: Crucial for the experimental determination of 3D band structure, the increase of λ_{PE} translates, via the Heisenberg uncertainty principle, to sharpening of the intrinsic resolution of the ARPES experiment in the out-of-plane momentum (k_z) which is defined as $\Delta k_z = \lambda_{PE}^{-1}$.

I disagree with this argument directly linking the mean free path length of the electron to the out-of-plane momentum resolution, as it is too simplistic a view. The authors' paper [ref28] is well considered and I partially agree with it, but I am concerned that this simplistic representation is in fact misleading to other researchers. I have seen papers confusing resolution with accuracy. The source of the initial state of the periodic k_z dispersion is from deeper than the surface region limited by the mean free path. The surface has the effect of scattering of the photoelectrons partially, increasing the structureless background intensity and broadening the peak. However, the k_z dispersion periodicity is not altered by the Heisenberg uncertainty [F.M. & S.S. PRB 105 (2022) 235126.].

2) the matching approach of LEED and surface structure

The main argument of this paper is the proposal to introduce end states using the LEED method. For the Ag example, a comparison with the calculated final state is made. No detailed surface structure information for samples and simulations. I would like the author to add more explanation.

Moreover, there are no detailed surface structure information for GaN and Si too.

For reconstructed surfaces and surfaces modified with molecular adsorbates, Umklapp scattering replica bands are often observed. This may be true for AlN/GaN, but not for Si(001)2x1 surfaces.

3) Non-FE effects beyond ARPES

This is also an interesting topic to discuss.

Regarding circular dichroism, I would like to draw the authors' attention to recent publications [P.K. & F.M. J. Electron Spectrosc. Relat. Phenom., 258, (2022) 147219] that discuss the importance of multiple scattering and itinerant band dispersion.

Reviewer #2 (Remarks to the Author):

Referee Comments

This paper describes how can be the final states of the ARPES at $h\nu=300\sim 1300\text{eV}$ in the case of Ag(001) at $\sim 12\text{ K}$. So far the occupied electronic states band dispersions have been mostly discussed by the analyses assuming the free electron band dispersions with inner potential V_0 of the unoccupied final electronic states except for the low $h\nu$ ARPES. However, the Fermi surface constant energy contours (CEC) around the $\bar{\Gamma}$ points are found here to be slightly modified depending upon the k_z values corresponding to the 2, 3, 4, 5th k_z Brillouin Zones (BZs). Although the results may not contribute to the essential analyses of the occupied valence band dispersions, the observed results may attract the curiosity of many ARPES scientists and theoreticians. This is because 3D band dispersions $E_B(k_x, k_y, k_z)$ are now desired to be experimentally measured by ARPES to discuss the complex physics in rather many materials for new physics and/or application.

The main conclusion of this paper is that multiband final states (MBFSs) must be considered in the case of Ag(100) ARPES in the above mentioned $h\nu$ region to better understand the observed detailed EF CEC. Noticeable partial broadening of the EF CEC is qualitatively explained by considering the

MBFSs and excessive broadening due to such effects. The authors thought as several MBFSs with different k_z s will contribute to the ARPES with E_k and $K_{||}$.

The details of the theoretical approaches are given in Supplementary Material, where ab-initio self-consistent electronic structure calculation is performed. The electronic structure of semi-infinite crystal was calculated to evaluate the self-consistent-field (SCF) within the relativistic multiple scattering approach using the Green's function KKR formalism in the tight binding mode. The SCF potential thus evaluated was then used for the ARPES calculation within one-step model. The final states corresponding to the time-reversed LEED states were obtained using the layer KKR technique.

Since this paper discussed the point which has not so far discussed in detail, it is worthwhile to be published. However, more details of experiments must be added before publication.

First, the experimental configurations, namely, photon incidence angle onto the sample surface, photoemission detection configuration must be clarified. Then readers would like to know whether the results in Fig.1(b) are measured by some rotation of the sample along the x direction or not. Namely, were the $k_x = \pi/1.5 \text{ \AA}^{-1}$ data measured without any motion of the sample?

Secondly the intensity distribution is much different between Fig.1(b) and Fig.S1. More convincing explanation is desired for publication.

Thirdly, the probing depth of the photoelectrons may be in most materials nearly 10 times different between the 300 eV and 1300 eV. Then the contribution of the surface electronic structures must be also checked and discussed in the report for discussing the delicate difference of the shape of FS CEC.

I hope that such discussions are included it is surely worthwhile to be published in this journal.

Reviewer #3 (Remarks to the Author):

The authors report observation of nontrivial electron band splitting and broadening in the angle-resolved photoemission spectra of sputter-annealed Ag(100) surface, acquired at the soft x-ray range. The effect is ascribed to the multiband final states (MBFSs) during the photoelectron ejection process. They use Green's function KKR method (which was also much utilized by the same authors in their pioneering works on this subject), and argue that the hybridization potential between the final state plane waves depends on the momentum separation of the plane waves, rather than their own kinetic energies (being high or low). They then generalize this to other (semiconducting) material families, posing challenges in simple generalization of the proposed explanation. I find this work of fundamental relevance and timely interest to the specialized field of photoemission spectroscopy, especially when the technique rapidly becomes more popularized (and certain necessary sophistication is getting lost in the process). Barring a few technical questions, I am happy

to recommend publication. However, as it stands now, the readability is much compromised by long sentences, broken up statements, compound words, and jargons.

Main questions/comments I have:

- can c-axis domains or stacking faults be comfortably ruled out over nm-scale in the sputter annealed Ag(100) sample? The k_z broadening and even band splitting should be differentiated from structural contributions.
- final states and Bloch waves should be better illustrated as a cartoon sketch in the early part of the manuscript. This can dramatically improve readability to general readers.
- What are the bands that bend towards zero energy near Γ and X in Figure 3?
- Is there a physical reason/argument, or any first principles support, that ΔV will be negative at small Δk , and switch sign at a "critical " Δk ?

Minor comments:

- Fig.S2 is hardly eligible even if zoomed in. It's such an informative and important figure that I'd even encourage the authors consider moving it to the main figures.
- On the contrary, Fig.4 seems very thin, and the information density is very low.
- The author is encouraged to break up long sentences, homogenize the language/terminology, and add lightweight formula where needed for self-contained-ness.
- The author should comment on the necessity and caveats associated with a grazing incident measurement geometry especially in the context of this study
- The discussion of XPD in the end is rather confusing. What purpose does that serve?
- what's the difference between p_{ph} and p_{hv} ? They are not defined clearly.

Response to the reviewers

We thank the referees for their careful reading and positive assessment of our work. Their constructive critique and suggestions have helped us improve its presentation in the ms. Below we answer the referees' remarks point by point.

Reviewer #1 (Remarks to the Author):

This report by Strocov et al. is an important work leading to an accurate understanding and precise analysis of photoelectron spectroscopy. This is because the authors attempt a detailed comparison with the free-electron approximate final state and the more realistic pseudopotential approximate final state, based on several experimental examples. Since the free electron approximation is only an approximation method, it is natural that the discrepancy between the actual and the approximation will eventually be exposed once a precise discussion begins, even in the high-energy region. In this sense, the title 'Is the approximation valid?' at first glance seems to already provide a 'no' answer, but it was interesting in that it gave both examples where the approximation breaks down (Ag and GaN) and where it does not (Si). In recommending this manuscript to your journal, I would like to make the following three comments. All of these points are of concern to me as a first-time reader, and I hope that the authors will use them to improve this manuscript.

1) relation of mean free path and k_z dispersion resolution

P2-I48: Crucial for the experimental determination of 3D band structure, the increase of λ_{PE} translates, via the Heisenberg uncertainty principle, to sharpening of the intrinsic resolution of the ARPES experiment in the out-of-plane momentum (k_z) which is defined as $\Delta k_z = \lambda_{PE}^{-1}$.

I disagree with this argument directly linking the mean free path length of the electron to the out-of-plane momentum resolution, as it is too simplistic a view. The authors' paper [ref28] is well considered and I partially agree with it, but I am concerned that this simplistic representation is in fact misleading to other researchers. I have seen papers confusing resolution with accuracy. The source of the initial state of the periodic k_z dispersion is from deeper than the surface region limited by the mean free path. The surface has the effect of scattering of the photoelectrons partially, increasing the structureless background intensity and broadening the peak. However, the k_z dispersion periodicity is not altered by the Heisenberg uncertainty [F.M. & S.S. PRB 105 (2022) 235126.].

> We thank the reviewer for this elucidating comment. The connection of λ_{PE} with the intrinsic Δk_z is based on a mathematical fact that the Fourier transform of a damped oscillator (in our case the final-state wavefunction damped within $1/2\lambda_{PE}$, with the factor 2 coming from squaring of the wavefunction for the electron density) is a Lorentzian having the fullwidth twice the damping constant (in our case $\Delta k_z = 1/\lambda_{PE}$). If this final state couples to an infinitely propagating initial-state Bloch wave having well-defined \mathbf{k} , this Δk_z is exactly the intrinsic resolution of the ARPES experiment in k_z . The reviewer is perfectly right, however, that this is a simplified view neglecting the surface effects. First, the final-state wavefunction does not terminate sharply at the surface, but continues to vacuum as a superposition of plane waves (including evanescent ones) that broadens the ideal Lorentzian distribution in k_z . Second, the initial-state Bloch wave does not propagate to vacuum, but decays to a superposition of evanescent plane waves that distorts its δ -function distribution in k_z . These surface effects have found their exhaustive theoretical description in the celebrated work by Feibelman and Eastman (PRB **10** (1974) 4932). They have found, however, that the shape of the function imposing the relaxed k_z conservation in the ARPES experiment stays reasonably close to the Lorentzian with the width $\Delta k_z = 1/\lambda_{PE}$, justifying our simplified approach. Admittedly, these insightful details have not been addressed at full in the first author's early paper referred to by the reviewer. Worth mentioning is also the surface photoemission caused by a change in the dielectric constant across the surface, whose interference with the bulk

photoemission results in broadening and asymmetry of the photoemission peaks (Miller et al, PRL **77** (1996) 1167, Hansen et al, PRB **55** (1997) 1871 and the follow-up works). However, all surface effects progressively reduce with increase of λ_{PE} towards high excitation energies, the focus of the present work. Therefore, their vivid discussion in the ms would be off-topic, in particular in view of the length limit of Nature Comm. In the revised version, we have only noted that our approach to the k_z conservation neglects the surface effects, and given a few references. The paper recommended by the reviewer is also cited in the context of relaxation of the non-symmorphic selection rules at the GaN(1000) surface.

Regarding the remaining comments at this point, we certainly agree that the periodicity of the k_z dispersion of the ARPES peaks comes from the initial-state propagation into the crystal much deeper than λ_{PE} (in fact, over the light absorption length) whereby its k_z is a well-defined quantum number. This periodicity actually reflects the periodicity of the initial-state Fourier components of the initial state at discrete values of $k_z + G_z$, where \mathbf{G} is a bulk reciprocal-lattice vector. We also agree that the intrinsic k_z resolution should not be confused with the intrinsic accuracy, with the latter reflecting the deviations of the ARPES peaks from the exact band energies. These deviations can occur, for example, due to sharp variation of the matrix element across Δk_z , or asymmetry of the density of states in the extremal points of the k_z dispersions. All these phenomena are discussed in detail in the cited papers of the first author in J. Electr. Spectr. and Relat. Phenom. **130** (2003) 65 and **229** (2018) 100, and are beyond the scope of the present paper.

2) the matching approach of LEED and surface structure

The main argument of this paper is the proposal to introduce end states using the LEED method. For the Ag example, a comparison with the calculated final state is made. No detailed surface structure information for samples and simulations. I would like the author to add more explanation.

Moreover, there are no detailed surface structure information for GaN and Si too.

For reconstructed surfaces and surfaces modified with molecular adsorbates, Umklapp scattering replica bands are often observed. This may be true for AlN/GaN, but not for Si(001)2x1 surfaces.

> Our Ag(100) samples were prepared by standard ion-sputtering/annealing cycles described in numerous previous works. LEED patterns have shown the expected (1x1) surface structure without trace of any superstructures. Neither have any previous works detected any superstructures on the Ag(100) prepared in the same way. The epitaxial GaN samples investigated in our work were capped by a 2-nm thick pseudomorphic AlN layer. The RHEED images acquired during their growth have demonstrated formation of clean monolayers of GaN without any superstructure. This interface, actually employed in GaN transistor heterostructures, has also been investigated in numerous previous TEM studies, for example, E. Schilirò *et al*, Nanomaterials **11** (2021) 3316; Li & Zhang, AIP Advances **13** (2023) 015214. No superstructures on the GaN surface have been identified as well. Surface of the Si(100) samples used in this work has been passivated by atomic H using the well-established procedure of etching in buffered HF solution, for entries see the literature in our Ref. 56. With the H atoms quenching the dangling bonds on the Si(100) surface, this procedure ensures the formation of the (1x1) surface without any superstructures as confirmed by numerous AFM, STM and electron-diffraction studies. Therefore, the experimental results reported in our work have not been impaired by any surface reconstructions. Because of the importance of this discussion, raised by the referee, we have added it to Methods of the revised ms. Furthermore, we have discussed and ruled out a possibility of out-of-plane charge-density waves that might in principle have also produced multiple ARPES structures.

3) Non-FE effects beyond ARPES

This is also an interesting topic to discuss.

Regarding circular dichroism, I would like to draw the authors' attention to recent publications [P.K. & F.M. J. Electron Spectrosc. Relat. Phenom., 258, (2022) 147219] that discuss the importance of multiple scattering and itinerant band dispersion.

> This is indeed a relevant work that we are citing in the revised version.

Reviewer #2 (Remarks to the Author):

Referee Comments

This paper describes how can be the final states of the ARPES at $\hbar=300\sim 1300\text{eV}$ in the case of Ag(001) at $\sim 12\text{ K}$. So far the occupied electronic states band dispersions have been mostly discussed by the analyses assuming the free electron band dispersions with inner potential V_0 of the unoccupied final electronic states except for the low \hbar ARPES. However, the Fermi surface constant energy contours (CEC) around the points are found here to be slightly modified depending upon the k_z values corresponding to the 2, 3, 4, 5th k_z Brillouin Zones (BZs). Although the results may not contribute to the essential analyses of the occupied valence band dispersions, the observed results may attract the curiosity of many ARPES scientists and theoreticians. This is because 3D band dispersions $E_B(k_x, k_y, k_z)$ are now desired to be experimentally measured by ARPES to discuss the complex physics in rather many materials for new physics and/or application.

The main conclusion of this paper is that multiband final states (MBFSs) must be considered in the case of Ag(100) ARPES in the above mentioned \hbar region to better understand the observed detailed EF CEC. Noticeable partial broadening of the EF CEC is qualitatively explained by considering the MBFSs and excessive broadening due to such effects. The authors thought as several MBFSs with different k_z s will contribute to the ARPES with E_k and K_{\parallel} .

The details of the theoretical approaches are given in Supplementary Material, where ab-initio self-consistent electronic structure calculation is performed. The electronic structure of semi-infinite crystal was calculated to evaluate the self-consistent-field (SCF) within the relativistic multiple scattering approach using the Green's function KKR formalism in the tight binding mode. The SCF potential thus evaluated was then used for the ARPES calculation within one-step model. The final states corresponding to the time-reversed LEED states were obtained using the layer KKR technique. Since this paper discussed the point which has not so far discussed in detail, it is worthwhile to be published. However, more details of experiments must be added before publication.

First, the experimental configurations, namely, photon incidence angle onto the sample surface, photoemission detection configuration must be clarified. Then readers would like to know whether the results in Fig.1(b) are measured by some rotation of the sample along the x direction or not. Namely, were the $k_x=1.5\text{ \AA}^{-1}$ data measured without any motion of the sample?

> The experimental configuration is presented in detail in the dedicated paper on our soft-X-ray ARPES facility (J. Synchr. Rad. **21** (2014) 32) that we reference in the Methods. However, following the reviewer's advice, in the Methods we have explicitly indicated the grazing-incidence angle and the analyzer slit orientation in the X-ray incidence plane. With this slit orientation, the results in Fig. 1 (b) did not involve any sample rotation.

Secondly the intensity distribution is much different between Fig.1(b) and Fig.S1. More convincing explanation is desired for publication.

> The difference in the intensity distribution between the experiment and calculations should have the same origins as the difference in the dispersions of the spectral structures, discussed in the Supplementary. In brief, two origins can be deciphered: (1) the increase of I_{max} to 5 was yet insufficient for high energies, whereas its further increase was beyond any realistic computational time; (2) the total-energy minimization used to generate the self-consistent potential might be quite insensitive to its

high-frequency Fourier components, critically affecting the high-energy electron states. In the revised ms we have mentioned the intensity differences, and added a short version of the above discussion to the Methods.

Thirdly, the probing depth of the photoelectrons may be in most materials nearly 10 times different between the 300 eV and 1300 eV. Then the contribution of the surface electronic structures must be also checked and discussed in the report for discussing the delicate difference of the shape of FS CEC. I hope that such discussions are included it is surely worthwhile to be published in this journal.

> Answering to the Reviewer 1, we have carefully analysed surface contributions to the ARPES spectra which might have been induced by potential surface reconstructions. This analysis has been partly included into the revised ms. In brief, all samples investigated in our work had unreconstructed (1x1) surfaces, excluding any significant distortions of the ARPES response compared to the bulk states. The intrinsic Shockley-Tamm surface states – such as the *d*-state derived surface state on Ag(100) seen in the experimental Fig. 3 (b) at E_B around -5.2 eV and calculated (e) at around -4.2 eV – are easily identified by their flat k_z dispersion, and do not interfere with our analysis of the k_z dispersive bulk states. Furthermore, all other known Shockley surface states of Ag(100) are located above the Fermi level. These states have been in the past studied by the inverse photoemission (see e.g. Himpsel et al., PRB **46** (1992) 9719 or Savio et al., Surf. Sci. **486** (2001) 65). Therefore, we can rule out any spectral structures related to these surface states in our experimental Fermi-surface or k_z dispersion maps.

Reviewer #3 (Remarks to the Author):

The authors report observation of nontrivial electron band splitting and broadening in the angle-resolved photoemission spectra of sputter-annealed Ag(100) surface, acquired at the soft x-ray range. The effect is ascribed to the multiband final states (MBFSs) during the photoelectron ejection process. They use Green's function KKR method (which was also much utilized by the same authors in their pioneering works on this subject), and argue that the hybridization potential between the final state plane waves depends on the momentum separation of the plane waves, rather than their own kinetic energies (being high or low). They then generalize this to other (semiconducting) material families, posing challenges in simple generalization of the proposed explanation. I find this work of fundamental relevance and timely interest to the specialized field of photoemission spectroscopy, especially when the technique rapidly becomes more popularized (and certain necessary sophistication is getting lost in the process). Barring a few technical questions, I am happy to recommend publication.

However, as it stands now, the readability is much compromised by long sentences, broken up statements, compound words, and jargons.

> Our two native English co-authors have re-read the ms. We hope that the introduced scientific language corrections have improved its readability.

Main questions/comments I have:

- can c-axis domains or stacking faults be comfortably ruled out over nm-scale in the sputter annealed Ag(100) sample? The k_z broadening and even band splitting should be differentiated from structural contributions.

> We have certainly checked the reproducibility of our results between a few sample preparation rounds of 3-5 sputtering-annealing cycles each. Moreover, we measured spectra in a few different locations over the sample. The characteristic spectral features due to the MBFS were identical. Furthermore, the c-axis domains or stacking faults, if existed, might be only aperiodic, producing merely broadening in the k_z direction rather than the separate branches of k_z dispersion observed in our experiment. We can therefore rule out any spectroscopic signatures of such structural defects even if they existed.

- final states and Bloch waves should be better illustrated as a cartoon sketch in the early part of the manuscript. This can dramatically improve readability to general readers.

> This is an extremely valuable suggestion. In the Introduction, we have added Fig. 1, the cartoon figure that illustrates the main idea of the multiband final states, and the corresponding explanation. We believe this has indeed improved the readability of the ms.

- What are the bands that bend towards zero energy near Gamma and X in Figure 3?

> Such bands bending towards zero energy are characteristic of the complex band structure as a function of complex k_z . Their origin can be explained starting from $V_i=0$ and the simplest free-electron dispersion which can be expressed as $E=(\mathbf{k}+\mathbf{G})^2$ (setting $\frac{\hbar^2}{2m}=1$). Assuming for simplicity that \mathbf{k} is along the z-axis (its k_{xy} component is fixed by the parallel momentum conservation in the photoemission process) and \mathbf{G} is in the (xy)-plane, we obtain $E=k_z^2+G_{xy}^2$, or $k_z=\sqrt{E-G_{xy}^2}$. If $E>G_{xy}^2$, we arrive at the conventional parabolic band dispersion as a function of $\text{Re}k_z$. If $E<G_{xy}^2$, k_z becomes purely imaginary, $k_z=i\sqrt{G_{xy}^2-E}$ with $\text{Re}k_z=0$. This is a vertical line falling from $E=G_{xy}^2$ vertically down along the energy axis. If we now introduce $V_i\neq 0$, the transition between the parabolic and vertical dispersion smoothens into the bands bending towards zero energy, as can be seen in our Fig. 4 (a). This pattern continues to the non-free-electron case in Fig. 4 (b). The reader may also be interested to look at the complex band structure dispersions as a function of $\text{Im}k_z$, now included in the Supplemental Material (2) as Fig. S1. Further discussion of characteristic features of the complex band structure and its difference from the conventional real- \mathbf{k} band structure can be found in the classical works in our Refs. 17 and 43-45.

- Is there a physical reason/argument, or any first principles support, that ΔV will be negative at small Δk , and switch sign at a "critical" Δk ?

> In the $\Delta\mathbf{k}=0$ limit, the $V_{\Delta\mathbf{k}}$ component of the pseudopotential $V(\mathbf{r})$ reduces to the integral of the latter over the unit cell, which has to be the largest negative. In the $\Delta\mathbf{k}\rightarrow\infty$ limit, $V(\mathbf{r})$ is integrated with an infinitely fast oscillating plane wave that sets $V_{\Delta\mathbf{k}}$ to zero asymptotically. Between these two extremes, strictly speaking, the sign does not have to change. However, it does so for the popular 'empty-core' type of pseudopotentials, where $V(\mathbf{r}) = 0$ inside the effective ion-core radius r_c and $\propto 1/r$ outside. In this case $V_{\Delta\mathbf{k}} \propto \cos(\Delta\mathbf{k}\cdot r_c)$, with the 'critical' $\Delta K = \pi/2r_c$ (see, for example, the latest review by Chelikowsky in Reference Module in Materials Science and Materials Engineering, Elsevier, 2022). This detail is not important for our analysis, though.

Minor comments:

- Fig.S2 is hardly eligible even if zoomed in. It's such an informative and important figure that I'd even encourage the authors consider moving it to the main figures.

> We have rearranged the panels Fig. S2 in order to make them eligible. This figure is logically attached to the calculations in Fig. S2, which is why we have left it in the Supplementary.

- On the contrary, Fig.4 seems very thin, and the information density is very low.

> We agree that Fig. 4 appears quite thin. We thought of different ways to integrate it into Fig. 1 or 2, but found that this would destroy the graphic coherence of the figures. However, we deem this figure important enough to be exposed separately. Indeed, such a vivid structure of three MDC peaks instead of the two expected from the conventional free-electron final states, all three having different widths

because of the MBFS effects, has never been reported before. Reproducing such a non-trivial spectral response may be an excellent benchmarking case for first-principles ARPES calculations.

- The author is encouraged to break up long sentences, homogenize the language/terminology, and add lightweight formula where needed for self-contained-ness.

> Our two native English co-authors have re-read the ms. We hope that the introduced scientific language corrections have improved its readability. A few essential formulas have been added. More formulas would leave the story patchy anyway, therefore, we have added a comprehensive Supplemental Material (2) compiling the formalism of the LEED matching approach in the context of its connection to the final-state band structure.

- The author should comment on the necessity and caveats associated with a grazing incident measurement geometry especially in the context of this study

> The advantages of the grazing-incidence experimental geometry are discussed in detail in the dedicated paper on our soft-X-ray ARPES facility (J. Synchr. Rad. **21** (2014) 32) that we reference in the Methods. Following the reviewer's advice, however, in the revised version we have indicated the associated increase of the photoelectron signal. If the sample is turned to the grazing incidence along the horizontal axis (to avoid blowing up of the larger horizontal light footprint on the sample) this experimental geometry does not hide any caveats, in particular for our large and homogeneous samples of Ag, GaN and Si.

- The discussion of XPD in the end is rather confusing. What purpose does that serve?

> Before submitting the ms, we had been discussing our findings with quite a few ARPES specialists. Surprisingly, many tended to confuse the non-free-electron effects in the photoemission final states with photoelectron diffraction. Surely, both phenomena involve multiple scattering of photoelectrons at their origin. Their crucial difference is, however, that the photoemission process produces an inherently coherent wavefield involving all atomic sites, whereas the photoelectron diffraction is an incoherent process where all photoelectrons scatter individually. We deem this crucial aspect important to highlight for the general reader.

- what's the difference between p_{ph} and p_{hv} ? They are not defined clearly.

> These two actually stand for the same thing, the photon momentum. We thank the reviewer for spotting the inconsistency, and have fixed it.

We thank the reviewers again for their constructive points of critique. We hope that the above answers and the revisions made to the ms exhaust the referees' concerns. Furthermore, we have added a Supplemental Material outlining details of our pseudopotential-based LEED calculations in the context of their connection to the final-state band structure. We hope that revised version fully stands the high scientific and presentation standards of Nature Communications.

REVIEWERS' COMMENTS

Reviewer #1 (Remarks to the Author):

I would like to thank the author for his extensive discussion of my comments.

It was also an opportunity to deepen my understanding.

All my concerns were answered and resolved.

I highly recommend publishing this version in Nature Communication.

Reviewer #2 (Remarks to the Author):

This paper discussed the reason why the ARPES of Ag(001) measured at photon energies between 300 and 1300 eV showed noticeable difference of the constant energy contours (CECs) at the Fermi level energy for different k_z corresponding to the n th CECs around $(k_x, k_y) = (0, 0)$ points.

In comparison with theoretical ARPES calculations, the difference is assigned to be resulting from the complex final states, which have been so far assumed to be free electron like. Since all details of experimental configuration, sample quality as well as theoretical details are given in this paper, I believe more detailed experiments may be performed in the near future and further check of the validity may be made on various materials.

As the initial work, this paper is worthwhile to be published in Nature Commun.

Reviewer #3 (Remarks to the Author):

The authors have addressed all my main concerns - thank you. Even though I still feel certain sentences are long and jargon-studded for a general audience, I recommend publication so long as the other reviewers find no issue with the readability.